# Severe Acute Respiratory Syndrome Coronavirus 2 (SARS-CoV-2) and *Bordetella bronchiseptica* Co-Infection in a Stem Cell Transplant Patient

**DOI:** 10.3390/antibiotics11091200

**Published:** 2022-09-05

**Authors:** Michael Pierce, Wendy Slipke, Mark Biagi

**Affiliations:** 1Department of Pharmacy, UW Health University of Hospital, Madison, WI 53792, USA; 2Department of Pharmacy, UW Health SwedishAmerican Hospital, Rockford, IL 61104, USA; 3College of Pharmacy, University of Illinois at Chicago, Rockford, IL 61107, USA

**Keywords:** COVID-19, SARS-CoV-2, *Bordetella bronchiseptica*, co-infection

## Abstract

*Bordetella bronchiseptica* infections may be overlooked by clinicians due to the uncommon encounter of this pathogen in humans and common isolation of co-pathogens. However, the isolation of *B. bronchiseptica* in immunocompromised individuals may represent a true infection. We report our experience with the fatal case of a stem cell transplant recipient, co-infected with SARS-CoV-2 and *B. bronchiseptica*, who was considered fully vaccinated (two doses) at the time of her case in spring 2021. Future studies are needed to evaluate the incidence of bacterial co-infections in immunosuppressed individuals with SARS-CoV-2 and clinicians should remain cognizant of the potential pathogenic role of uncommon pathogens isolated in these individuals.

## 1. Introduction

Viral respiratory infections have been historically associated with an increased risk of bacterial co-infections, including secondary and superinfections. Current literature suggests lower rates of bacterial co-infections among patients infected with SARS-CoV-2 disease (COVID-19) compared to previous influenza pandemics [1]. In a recent meta-analysis, bacterial co-infections and secondary bacterial infections were identified in 3.5% and 14.3% of COVID-19 patients, respectively [2]. However, the risk of bacterial infections with COVID-19 is less well understood among immunocompromised patients, including those with a transplant history [3]. In a previous study, the reported mortality rate among solid-organ transplant recipients infected with COVID-19 was 18%; however, rates of secondary and co-infections were not described [4]. In a separate study, bacterial or fungal co-infections were reported in 14/66 (21.2%) of solid-organ transplant recipients diagnosed with COVID-19 [5]. The clinical outcomes of stem cell transplant patients with COVID-19 are even more scarce, but based on available data, the case fatality rate among this population is 27% [6]. Studies describing the rate of secondary and co-infections among stem cell transplant recipients with COVID-19 are virtually non-existent.

Based on the available literature, immunocompromised individuals appear to be at higher risk of developing severe COVID-19 disease as well as co-infections compared to the general population. Underlying immunosuppression may further predispose these individuals to opportunistic pathogens not routinely encountered in clinical practice. Historically, *Bordetella bronchiseptica* infections in humans have been rarely reported [7]. In both animals and humans, the respiratory tract is the predominant site of *B. bronchiseptica* infections. In humans, the clinical significance of *B. bronchiseptica* isolation from the respiratory tract varies between individuals as the disease pattern ranges from asymptomatic colonization to severe pneumonia [8]. However, anecdotal data suggest a potential pathogenic role of *B. bronchiseptica* in the respiratory tract of immunocompromised patients. We report the fatal case of a fully vaccinated stem cell transplant recipient co-infected with COVID-19 and *B. bronchiseptica*. 

## 2. Case Presentation

A 50-year-old female patient presented to the emergency department (ED) in spring 2021 for new-onset worsening shortness of breath over the previous 24 h and a chronic cough that had started a few weeks earlier. Of note, she received levofloxacin (500 mg orally once daily for 7 days) approximately one month earlier for sinusitis complicated by a cough with green and yellow sputum production. She was also seen previously in our ED ten days earlier for abdominal pain and increasing stool output in her colostomy but reported no respiratory symptoms and was afebrile. In January 2021, a diverting descending loop colostomy was placed due to a non-healing perineal wound for over a year that had progressed to the point of causing complete loss on the left side of her sphincter complex. Past medical history was significant for Hodgkin’s lymphoma (diagnosed August 2017), allogenic hematopoietic stem cell transplant (July 2020), hypothyroidism, depression with anxiety, and gastroesophageal reflux disease. She was receiving nivolumab (240 mg intravenous every 14 days) for refractory Hodgkin’s lymphoma. She had received her first and second COVID-19 vaccinations approximately 3 and 6 weeks prior to her presentation to the ED, respectively, and her most recent tetanus/diphtheria/acellular pertussis (Tdap) vaccination in November 2014. She lived at home with her parents and pet cat. 

On physical examination day of presentation (day 0), the patient’s weight was 108.9 kg with a BMI of 39.94 kg/m^2^, temperature 36.7 °C, respiratory rate 32 breaths per minute, blood pressure 114/80 mm/Hg, and heart rate 109 beats per minute. Supplemental oxygen was administered via nasal cannula (6 L/min) due to an initial oxygen saturation of 68% on room air. A chest X-ray obtained in the ED demonstrated extensive bilateral, widespread, mixed ground-glass opacities (Figure 1A). Baseline laboratory values revealed elevated C-reactive protein (18.1 mg/dL), erythrocyte sedimentation rate (61 mm/h), and ferritin (727.3 ng/mL). White blood cell count (7.0 × 10^3^/µL) was within normal limits and did not demonstrate lymphopenia (25.6%). Baseline D-dimer was 256 ng/mL. A nasopharyngeal polymerase chain reaction swab was positive for COVID-19. The patient was transferred to the intensive care unit (ICU) where she was subsequently mechanically ventilated later on day 0 due to persistently increasing oxygen requirements and respiratory failure following admission.

In the ICU, the patient received tocilizumab (800 mg intravenous once) on day 0, remdesivir (200 mg intravenous once followed by 100 mg intravenous once daily) from days 0–9, and dexamethasone (4 to 10 mg intravenous once daily) from days 0–10. On day 4, a repeat chest X-ray demonstrated slightly increased bilateral opacities raising the concern for a potential superimposed bacterial co-infection. An endotracheal sputum culture was collected on day 5, and the patient was started on cefepime (2 g intravenous every 8 h) plus vancomycin (2 g intravenous loading dose followed by 1 g intravenous every 12 h). The endotracheal sputum culture resulted on day 7 positive for 2+ *B. bronchiseptica* identified by matrix-assisted laser desorption/ionization time-of-flight mass spectrometry. Vancomycin therapy was subsequently discontinued while the patient remained on cefepime. Levofloxacin (750 mg intravenous every 24 h) was started on day 7. A second endotracheal sputum culture collected on day 8 was positive for 1+ *B. bronchiseptica*. Due to the growth of *B. bronchiseptica* in two consecutive cultures the organism was sent to a reference laboratory for susceptibility testing (Table 1). From days 10–11, the patient developed leukocytosis (12.5 × 10^3^/µL on day 10 to 36.4 × 10^3^/µL on day 11) and a fever (38.1 °C). On day 11, a chest X-ray demonstrated improvement in the right lower lobe opacity and no change in the lower left lobe opacity (Figure 1B). Cultures were collected from multiple sites on day 11. Blood and urine cultures were negative, and an endotracheal sputum culture was positive for normal respiratory flora only. Additionally, on day 11, the patient had an acute decrease in hemoglobin (8.7 g/dL on day 10 to 5.2 g/dL on day 11) causing concern for an internal bleed. A computed tomography scan revealed a massive right retroperitoneal hematoma with smaller hematomas in the right psoas muscle. Despite aggressive measures, the patient continued to deteriorate over the following 24 h. Comfort care measures were initiated on day 12, and the patient expired later that day. The ultimate cause of death was presumed to be related to the patient’s retroperitoneal hemorrhage.

## 3. Discussion

Human infections caused by *B. bronchiseptica* have been rarely reported in the literature [7,8]. A potential explanation for the rarity of human *B. bronchiseptica* infections is that cross-immunity may be conferred by *B. pertussis* vaccination, although immunocompromised individuals may remain susceptible due to a diminished immune response [9]. The pathogenicity of *B. bronchiseptica* in humans is not well established but should be considered as a true clinical pathogen when isolated from the lower respiratory tract of immunocompromised patients [8,10]. While we acknowledge the possibility that *B. bronchiseptica* may have represented airway colonization in our patient, her immunosuppressed status and clinical presentation suggest that it played a pathogenic role. For example, our patient’s chronic cough began approximately a month before she developed shortness of breath, an important consideration given that chronic cough has been previously associated with human *B. bronchiseptica* infections [11,12]. Additionally, our case represents at least the second patient diagnosed with *B. bronchiseptica* pneumonia while receiving nivolumab [13] and the fourth documented patient co-infected with COVID-19 and *B. bronchiseptica* [14,15,16]. Clinical characteristics of patients co-infected with COVID-19 and *B. bronchiseptica* are summarized in Table 2.

To date, clinical trials for vaccines against COVID-19 have excluded severely immunocompromised patients [17,18,19]. Current COVID-19 treatment guidelines from the National Institutes of Health (NIH) recommend vaccination for HSCT patients but acknowledge that response rates may be attenuated in patients who are moderately to severely immunocompromised [20]. It is generally accepted that immunity to COVID-19 is developed two weeks after the second dose of a two-dose vaccination. In our case, the patient’s self-reported onset of shortness of breath was 19 days after receiving her second COVID-19 vaccination. The typical time to symptom onset of COVID-19 is ~5 days but may be delayed up to two weeks, raising the possibility that our patient could have contracted COVID-19 before developing full immunity following her second vaccine dose [21,22]. 

Our experiences reported here highlight a few key points for clinicians. First, clinicians should be cognizant of opportunistic infections, including those caused by rare pathogens, in immunosuppressed patients co-infected with COVID-19. In our case, it was felt that pathogen-directed therapy against *B. bronchiseptica* was warranted given her chronic cough, worsening respiratory status, and immunocompromised status due to both her baseline medical history and additional immunosuppressive effects of dexamethasone and tocilizumab. Second, a thorough history of present illness is imperative as demonstrated by our patient’s history of chronic cough and contact with a cat, a known risk factor for *B. bronchiseptica* infection [8,11]. Although susceptibility testing was not performed in our case for cefepime, multiple reports have demonstrated resistance against *B. bronchiseptica* [13,23,24] raising the possibility that empiric therapy may have been optimized earlier in the treatment course for our case. Third, for clinicians at sites unable to perform accurate species identification and/or susceptibility testing of uncommon organisms, testing at a reference laboratory is recommended. In our case, susceptibility testing was performed on the second positive culture for *B. bronchiseptica,* and results were not available until the day the patient expired. From an infectious standpoint, antimicrobial therapy could theoretically have been optimized earlier in the treatment course had the initial isolate from the first positive culture been referred to the reference laboratory. Finally, while vaccination is recommended for immunosuppressed patients, clinicians should counsel patients that efficacy data in this patient population is not well understood, and patients should consider continuing to practice social distancing and mask wearing. Future studies evaluating the incidence of co-infections and vaccine efficacy among COVID-19 patients with underlying immunosuppression are needed.

## Figures and Tables

**Figure 1 antibiotics-11-01200-f001:**
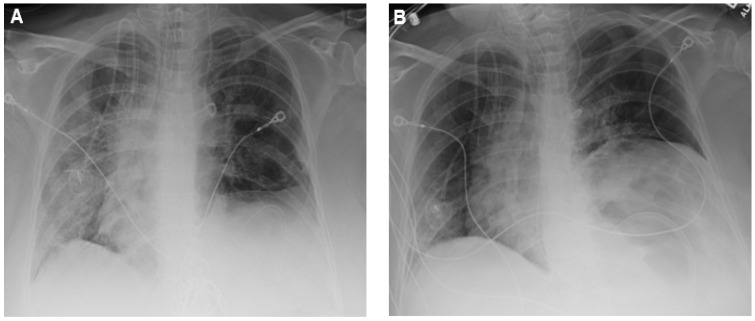
Chest X-rays showing (**A**) baseline bilateral infiltrates on day 0 and (**B**) mild improvement in the patchy airspace opacity in the right lower lobe and stable patchy opacity in the left lower lobe on day 11.

**Table 1 antibiotics-11-01200-t001:** Minimum inhibitory concentration (mg/L) and susceptibility interpretation of agents tested against *Bordetella bronchiseptica* isolated on day 8.

ATM	CAZ	CIP	IMI	MIN	P/T	T/C	TMP/SMZ
≥64 (R)	16 (I)	1 (S)	1 (S)	0.5 (S)	≤1/4 (S)	8/2 (S)	4/76 (R)

ATM = aztreonam, CAZ = ceftazidime, CIP = ciprofloxacin, IMI = imipenem, MIN = minocycline, P/T = piperacillin/tazobactam, T/C = ticarcillin/clavulanate, TMP/SMZ = trimethoprim/sulfamethoxazole, S = susceptible, I = intermediate, R = resistant.

**Table 2 antibiotics-11-01200-t002:** Clinical characteristics of patients with COVID-19 and *Bordetella bronchiseptica* co-infection.

	Age, Sex	Specimen	Underlying Conditions	Animal Exposure	Treatment	Outcome
Faqihi et al. [14]	30, M	Endotracheal aspirate	Idiopathic non-cystic bronchiectasis, vitamin D_3_ deficiency	Dog	Doxycycline	Cured
Nagarakanti et al. [15]	48, M	Sputum	Renal transplant, chronic obstructive pulmonary diseases, hypertension, diabetes mellitus, obesity, gout, and obstructive sleep apnea	None	Azithromycin Piperacillin/tazobactam	Cured
Papantoniou et al. [16]	47, M	Blood	None	Not reported	Piperacillin/tazobactam Meropenem	Cured
Pierce et al.	50, F	Endotracheal aspirate	Hodgkin’s lymphoma, allogenic hematopoietic stem cell transplant, hypothyroidism, depression with anxiety, and gastroesophageal reflux disease	Cat	Levofloxacin Cefepime	Died

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
