# Peer review of "Severe Acute Respiratory Syndrome Coronavirus 2 (SARS-CoV-2) and Bordetella bronchiseptica Co-Infection in a Stem Cell Transplant Patient"

_antibiotics, 2022, doi:10.3390/antibiotics11091200_

Round 1

Reviewer 1 Report

In this case report, the authors present the fourth documented patient coinfected with COVID-19 and B. bronchiseptica. The work is well structured and correctly written. This makes it interesting and easy to read. Authors should review the following points:

1. Please consider adding information in the introduction. It would be interesting if the authors briefly delve into bacterial co-infections in immunocompromised patients with COVID-19. Information on Bordetella bronchiseptica infections (detection, symptoms, consequences, etc.) should also be included.

2. The tables and figures with their respective legends were not included in the manuscript. Please insert your graphics (schemes, figures, etc.) in the main text after the paragraph of its first citation following the instructions for authors.

Author Response

In this case report, the authors present the fourth documented patient coinfected with COVID-19 and B. bronchiseptica. The work is well structured and correctly written. This makes it interesting and easy to read. Authors should review the following points:

  1. Please consider adding information in the introduction. It would be interesting if the authors briefly delve into bacterial co-infections in immunocompromised patients with COVID-19. Information on Bordetella bronchisepticainfections (detection, symptoms, consequences, etc.) should also be included.

Response: The introduction has been expanded to include information regarding co-infections in both the general population as well as special immunocompromised populations. Additional information regarding B. bronchiseptica specifically has been added to the introduction as well.

  1. The tables and figures with their respective legends were not included in the manuscript. Please insert your graphics (schemes, figures, etc.) in the main text after the paragraph of its first citation following the instructions for authors.

Response: All figures and tables have been added to the main text.

Reviewer 2 Report

Severe Acute Respiratory Syndrome Coronavirus 2 (SARS-CoV-2) and Bordetella bronchiseptica Co-infection in a Stem Cell Transplant Patient

Abstract section:

Line 13: Change the term "rarity" to "uncommon"

Line 16: quantify fully vaccinated: received 2 doses plus boosters?

Introduction section:

Lines 22-24: To provide a better and more up-to-date overview of bacterial co-infections in patients with SARS-CoV-2.

 Case pretetation section:

Line 33: Define the most exact time to classify it as chronic cough.

Line 36: For what reason and since when was the patient a carrier of a colostomy?

Line 69: Were hospital-acquired infections checked?

Discussion section:

The possibility of bacterial colonization of Bordetella should be added to the discussion.

Author Response

Severe Acute Respiratory Syndrome Coronavirus 2 (SARS-CoV-2) and Bordetella bronchiseptica Co-infection in a Stem Cell Transplant Patient

Abstract section:

Line 13: Change the term "rarity" to "uncommon"

            Response: Line 13 has been updated as recommended.

Line 16: quantify fully vaccinated: received 2 doses plus boosters?

Response: Lines 17-18 have been updated to provide clarity that the patient received two doses in Spring 2021 (prior to when boosters were recommended).

Introduction section:

Lines 22-24: To provide a better and more up-to-date overview of bacterial co-infections in patients with SARS-CoV-2.

            Response: The introduction has been updated as requested by Reviewers 1 and 2.  

Case presentation section:

Line 33: Define the most exact time to classify it as chronic cough.

Response: The most exact time documented in the electronic medical record is “few weeks earlier” per the patient. We are unable to provide a more exact time.

Line 36: For what reason and since when was the patient a carrier of a colostomy?

Response: We have added details to the main text that the patient had colostomy placement in January 2021 for a non-healing perineal wound.

Line 69: Were hospital-acquired infections checked?

Response: Only a sputum culture was collected on day 8. All cultures (blood, sputum, and urine) collected during the patient’s admission are reported within the case presentation.

Discussion section:

The possibility of bacterial colonization of Bordetella should be added to the discussion.

            Response: This has been incorporated into the first paragraph of the discussion.
